META-RESEARCH

# Evaluating the impact of open access policies on research institutions

**Abstract** The proportion of research outputs published in open access journals or made available on other freely-accessible platforms has increased over the past two decades, driven largely by funder mandates, institutional policies, grass-roots advocacy, and changing attitudes in the research community. However, the relative effectiveness of these different interventions has remained largely unexplored. Here we present a robust, transparent and updateable method for analysing how these interventions affect the open access performance of individual institutes. We studied 1,207 institutions from across the world, and found that, in 2017, the top-performing universities published around 80–90% of their research open access. The analysis also showed that publisher-mediated (gold) open access was popular in Latin American and African universities, whereas the growth of open access in Europe and North America has mostly been driven by repositories.

**CHUN-KAI (KARL) HUANG\*, CAMERON NEYLON, RICHARD HOSKING, LUCY MONTGOMERY, KATIE S WILSON, ALKIM OZAYGEN AND CHLOE BROOKES-KENWORTHY**

## Introduction

While there is substantial disagreement on the best route to achieve open access, the idea that research outputs should be freely available is broadly shared. Over the past decade, there has been a large increase in the volume of publications available open access (*Piwowar et al., 2018*), and this looks set to continue: indeed, a recent projection estimates that by 2025, 44% of all journal articles will be available as open access (OA) and that 70% of article views will be to OA articles (*Piwowar et al., 2019*).

This massive increase has largely been driven by policy initiatives. Medical research funders in the UK, such as the Wellcome Trust and Medical Research Council, and the National Institutes of Health in the US have led a wide range of funder policy interventions. Universities such as Harvard University, University of Liège, University of Southampton and others developed local polices and infrastructures that became more widely adopted. In 2018, a coalition of funders set out an initiative called Plan S that requires all

scholarly publications funded by public grants to be made immediately open access. This is the most ambitious, and therefore the most controversial, policy initiative to date with questions raised about the approach (*Rabesandratana, 2019*; *Haug, 2019*; *Barbour and Nicholls, 2019*), implementation details (*McNutt, 2019*; *Gómez-Fernández, 2019*; *Brainard, 2019*; *Agustini and Berk, 2019*), and unintended side effects for existing programs outside North America and North-western Europe (*Debat and Babini, 2019*; *Aguado-López and Becerril-García, 2019*).

A recent report showed a link between the monitoring of policy and its effectiveness, demonstrating that research outputs supported by funders that implemented monitoring and compliance checks for their policies were more likely to be published open access (*Larivière and Sugimoto, 2018*). By comparison, open access for works funded by Canadian funders, which did not monitor compliance, were shown to lag

**\*For correspondence:** karl. huang@curtin.edu.au

**Competing interests:** The authors declare that no competing interests exist.

substantially even when disciplinary effects were taken into account.

There is a need for critical and inclusive evaluation of open access performance that can address regional and political differences. For example, the SciELO project has successfully implemented an electronic publishing model for journals resulting in a surge of publisher-mediated open access (*Packer, 2009*; *Wang et al., 2018*). Recent work showed that, for biomedical research, there was a greater level of open access for articles published from countries with a lower GDP, particularly for those in sub-Saharan Africa (*Iyandemye and Thomas, 2019*). This provides evidence of national or regional effects on publication cultures that lead to open access. Meanwhile, another study showed that, for the field of Global Health, lower-ranked institutions are more likely to publish in closed outlets (*Siler et al., 2018*). They suggest this is due to the cost of article processing charges showing the importance of considering institutional context when examining open access performance.

Despite the scale and success (at least in some areas) of policy interventions, there is limited comparative and quantitative research about which policy interventions have been the most successful. In part this is due to a historical lack of high-quality data on open access, the heterogeneous nature of the global scholarly publishing endeavour, and the consequent lack of any baseline against which to make comparisons.

### Aim of Study

We have argued (*Montgomery et al., 2018*) that the key to understanding and guiding the cultural changes that underpin a transition to openness is analysis at the level of research institutions. While funders, national governments, and research communities create the environments in which researchers operate, it is within their professional spaces that choices around communication, and their links to career progression and job security are strongest. Analysis of how external policy leads to change at the level of universities is critical. However, providing accurate and reliable data on open access at the university level is a challenge.

The most comprehensive work on open access at the university level currently available is that included in the CWTS Leiden Ranking (*Robinson-Garcia et al., 2019*; *Robinson-Garcia et al., 2020*). This utilises an internal Web of Science database and data from Unpaywall to provide estimates of open access over a range of timeframes. These data have highlighted the broad effects of funder policies (notably the performance of UK universities in response to national policies) while also providing standout examples from regions that are less expected (for instance Bilkent University in Turkey).

A concern in any university evaluation is the existing disciplinary bias in large bibliographic sources used to support rankings. For example, the coverages of Web of Science and Scopus were shown to be biased toward the sciences and the English language (*Mongeon and Paul-Hus, 2016*). If we are to make valid comparisons of universities across countries, regions and funders to examine the effectiveness of open access policy implementation there is a critical need for evaluation frameworks that provide fair, inclusive and relevant measurement of open access performance.

Alongside coverage of data sources are issues of scope (which institutions, what set of objects), metrics (numbers or proportions) and data completeness. Our pragmatic assessment is that any evaluation framework should be tied to explicit policy goals and be shaped to deliver that. Following from our work on open knowledge institutions (*Montgomery et al., 2018*) our goals in conducting an evaluation exercise and developing the framework are as follows: (i) Maximise the amount of research content that is accessible to the widest range of users, focusing on existing formal research content for which metadata quality is sufficiently high to enable analysis; (ii) Develop an evaluation framework that drives an elevation of open access and open science issues to a strategic issue for all research-intensive universities; (iii) Develop a framework that is sensitive to and can support universities taking a range of approaches and routes towards delivering on these goals.

In terms of a pragmatic approach to delivering these goals we intend to:

1. Focus on research-intensive institutions, using existing rankings as a sample set.
2. Seek to maximise the set of objects which we can collect and track while connecting them to institutions (i.e., increase recall but not at the expense of precision).

3. Focus on proportions of open access as a performance indicator rather than absolute numbers.
4. Publicly report on the details of performance for high performing institutions (and provide strategic data on request to others).
5. Report on the diversity of paths being taken to deliver overall access by a diverse group of universities.
6. Develop methodology that is capable of identifying which policy interventions have made a difference to outcome measures and any 'signature' of those effects.

With the above in mind, this study proposes a set of requirements for evaluating open access performance at the institutional level, and presents a large-scale analysis of universities by drawing and integrating data from multiple data sources. This work differs from the CWTS Leiden Rankings by extending the coverage of research outputs beyond the Web of Science. The data workflow we have developed is also transparent, reproducible, and updateable, which makes robust and longitudinal analysis more easily attainable. We emphasise that a simple numerical ranking of universities cannot be justified given there is minimal significant difference across them. Instead, we highlight how the resulting comprehensive overview of the open access landscape and the underlying trends over time can provide deep insights on effects of policy interventions.

## Methods

To map the rate and degree of progress to open access, we developed a reproducible workflow capable of quantifying a wide range of open access characteristics at the institutional level. The overall workflow is summarised diagrammatically in *Figure 1*. This includes mapping open access definitions and the Unpaywall information we used to construct them. Briefly, we gather output metadata from searches in Microsoft Academic (*Sinha et al., 2015*; *Wang et al., 2019*), Web of Science and Scopus, for each university. From this full set we gather the corresponding Crossref DOIs from the metadata of each output focusing on this set. Unpaywall is consulted to determine open access status. Detailed discussions of the data sources, precise data snapshots used, and technical details of the data infrastructure can be found in *Supplementary file 1*. The code used in the

workflow is available via Zenodo at the following link.

We have decided to focus mainly on total open access, publisher-mediated open access (i.e., gold), and repository-mediated open access (i.e., green) due to the ease of comprehension, data quality, and ability to show potential effects of selected policies (*Table 1*). Levels of hybrid open access and green in home repository for selected universities (ones for which we have more confidence in the data) are also included to further support the analysis of policy effects.

As we have noted previously (*Huang et al., 2020a*), there is a sensitivity associated to the choices in bibliographic data sources when they are used to create a ranking. For this analysis we therefore chose to combine all three datasets: Microsoft Academic, Web of Science and Scopus. In the companion white paper (*Huang et al., 2020b*) we provide a comprehensive sensitivity analysis on the use of these different datasets, the use of different versions of Unpaywall, and the relations between confidence levels and sample size.

Briefly, it is our view that to provide a robust assessment of open access performance the following set of essential requirements must be met:

1. The set of outputs included in each category and a traceable description of how they were collected must be transparently described. Provided here by a description of the data sources and the procedures used to collect DOIs for each institution (*Supplementary file 1*). In this article, institutions define the categories of outputs but they could also be categorised by individuals, disciplines, or countries etc.
2. A clearly defined, open and auditable data source on open access status. Provided here by a defined and identified Unpaywall snapshot (*Supplementary file 1*).
3. A clearly defined and implementable description of how open access status data is interpreted in the form of the SQL query used to establish open access status categories for each DOI (*Figure 1* and *Supplementary file 1*). We decided to include the checkmark 'is_oa = TRUE' in the description of Bronze open access as this makes a slight difference to the numbers obtained from Unpaywall.
4. Provision of derived data and analysis in auditable form. Provided here the derived data as open data (*Huang et al., 2020c*), code for the analysis of derived data as Jupyter notebooks (*Huang et al., 2020d*),

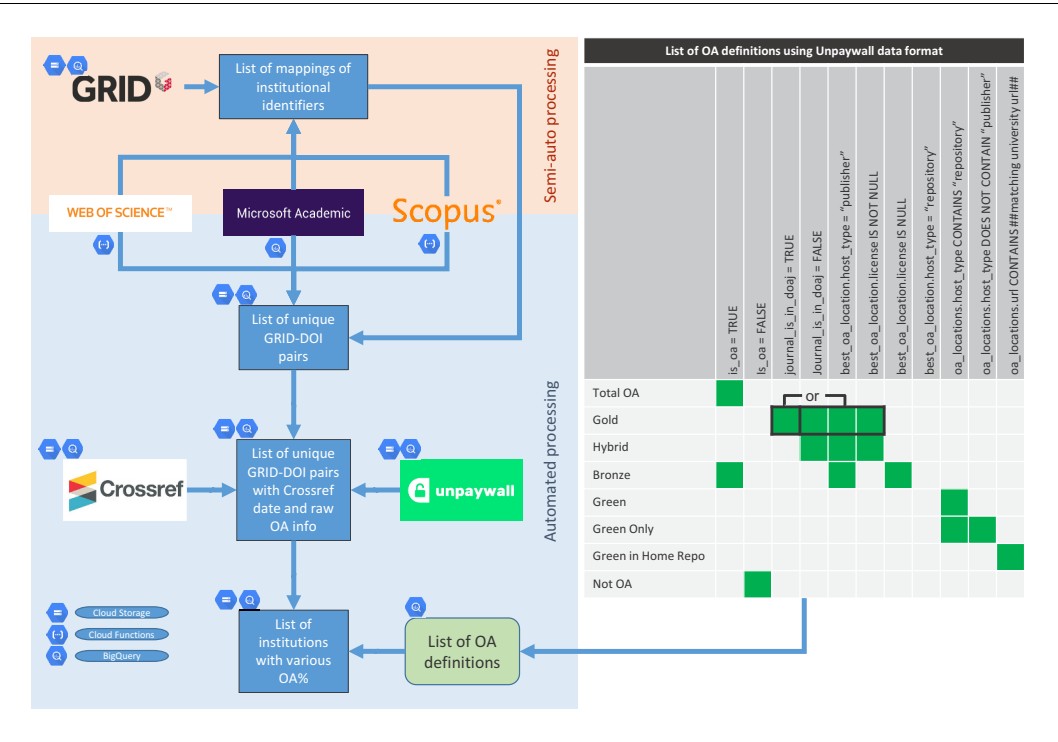

**Figure 1.** Analysis workflow. Diagrammatic summary of how data is collected and mapped against open access definitions using information from Unpaywall metadata.

and upstream data analysis in the form of SQL queries used (*Huang et al., 2020c*).

We have limited our data sharing in two ways. Firstly, we do not provide the full list of DOIs obtained from each source, due to Terms of Service restrictions. Secondly, we have not identified institutions individually except for those that fall within the top 100 globally for total open access, publisher-mediated open access, or repository-mediated open access. Both derived datasets of the de-anonymised top 100 and the full dataset containing all institutions in anonymised form are made available (*Huang et al., 2020c*).

## Results

### Global universities in terms of total open access, gold open access and green open access

In *Figure 2*, we present the open access performance of universities in different regions for the categories of total open access, publisher-mediated open access ('gold') and repository-mediated open access ('green') for publications assigned to the year 2017 (see *Figure 2—figure supplement 1* for equivalent plots for 2016 and

2018). We have chosen to focus on 2017 as this is the most recent year for which we have the most confidence on the completeness of data, taking into consideration the data collection process and issues surrounding embargoes. However, we do observe consistent general patterns across results for all three years. The top 100 institutions in each of the open access categories (for 2017) are also provided in *Figure 2—figure supplement 2*. This is, to our knowledge, the first set of university rankings that provides a confidence interval on the quantitative variable being ranked and compensates for the multiple comparisons effect (see *Supplementary file 1* for details). Across this top 100, the statistical difference between universities at the 95% confidence shows that a simple numerical ranking cannot be justified.

The high performance of a number of Latin American and African universities, together with a number of Indonesian universities, particularly with respect to gold open access, is striking. For Latin America this is sensitive to our use of Microsoft Academic as a data source, showing the importance of an inclusive approach. The outcomes for Indonesian universities are also consistent with the latest report on country-level analysis (*Van Noorden, 2019*). These suggest

**Table 1.** Definitions of open access.

Summary of different types of open access (OA) used in scholarly publishing. These definitions are not always mutually exclusive. For example, an article can be both Gold OA and Green OA. However, articles that are Green Only do not have any common element with articles classified as Gold OA by definition. This study focuses on the following categories: Total OA, Gold, Hybrid, Green and Green in Home Repo. Further discussions on open access definitions can be found in *Supplementary file 1*.

| OA type | Description |
| --- | --- |
| Total OA | A research output that is free to read online, either via the publisher website or in an OA repository. |
| Gold | A research output that is either published in a journal listed by the Directory of Open Access Journals (DOAJ), or (if journal not in DOAJ) is free to read via publisher with any license. |
| Hybrid | A research output that is published in a journal not listed by DOAJ, but is free to read from publisher with any license. |
| Bronze | A research output that is free to read online via publisher without a license. |
| Green | A research output that is free to read online via an OA repository. |
| Green Only | A research output that is free to read online via an OA repository, but is not available for free via the publisher. |
| Green in Home Repo | A research output that is free to read online via the matched affiliation's institutional repository. |

that the narrative of Europe and the USA driving a publishing-dominated approach to open access misses a substantial part of the full global picture.

The highest performers in terms of open access via repositories are dominated by UK universities. This is not surprising given the power of the open access mandate associated with the Research Excellence Framework to drive university behaviour. It is perhaps interesting that few US universities appear in this group (with Cal-Tech and MIT the exceptions). This suggests that while the National Institutes of Health

mandate has been very effective at driving open access to the biomedical literature, limited inroads have been made into other disciplines in the US context, despite the White House memorandum. As was seen in the Leiden Ranking, Bilkent University from Turkey also emerges as a standout performer.

### The global picture and its evolution

The levels of total open access, publisher-mediated open access and repository-mediated open access for 1,207 universities for publications in 2017 was also grouped by country (*Figure 2—*

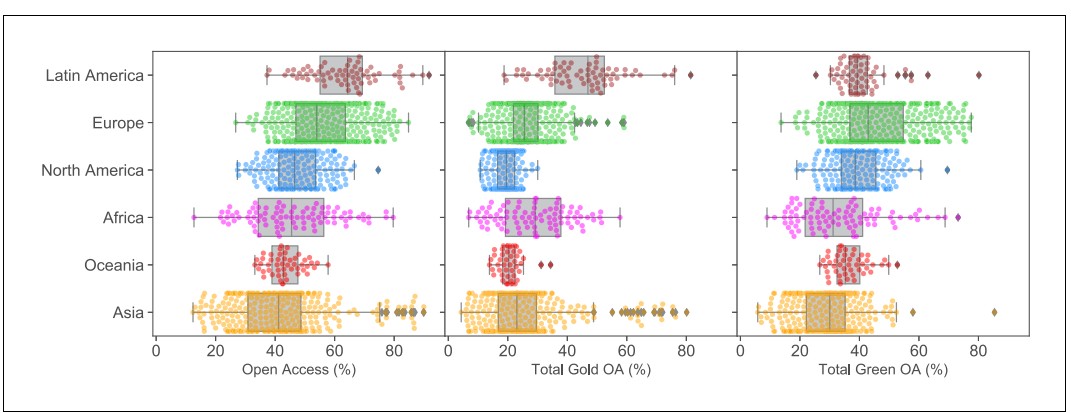

**Figure 2.** Open access performance of different geographical regions. Percentages of institutional Total OA, Gold OA and Green OA (left to right) grouped by regions for 2017. Parallel figures for 2016 and 2018 are provided in *Figure 2—figure supplement 1*.

The online version of this article includes the following figure supplement(s) for figure 2:

**Figure supplement 1.** Open access performance of different regions in 2016 and 2018.

**Figure supplement 2.** Top 100 universities in terms of performance in total open access, publisher-mediated open access (gold OA) and repository-mediated open access (green OA) for 2017.

**Figure supplement 3.** Percentage of institutional Total OA, Gold OA and Green OA (left to right) grouped by country for 2017.

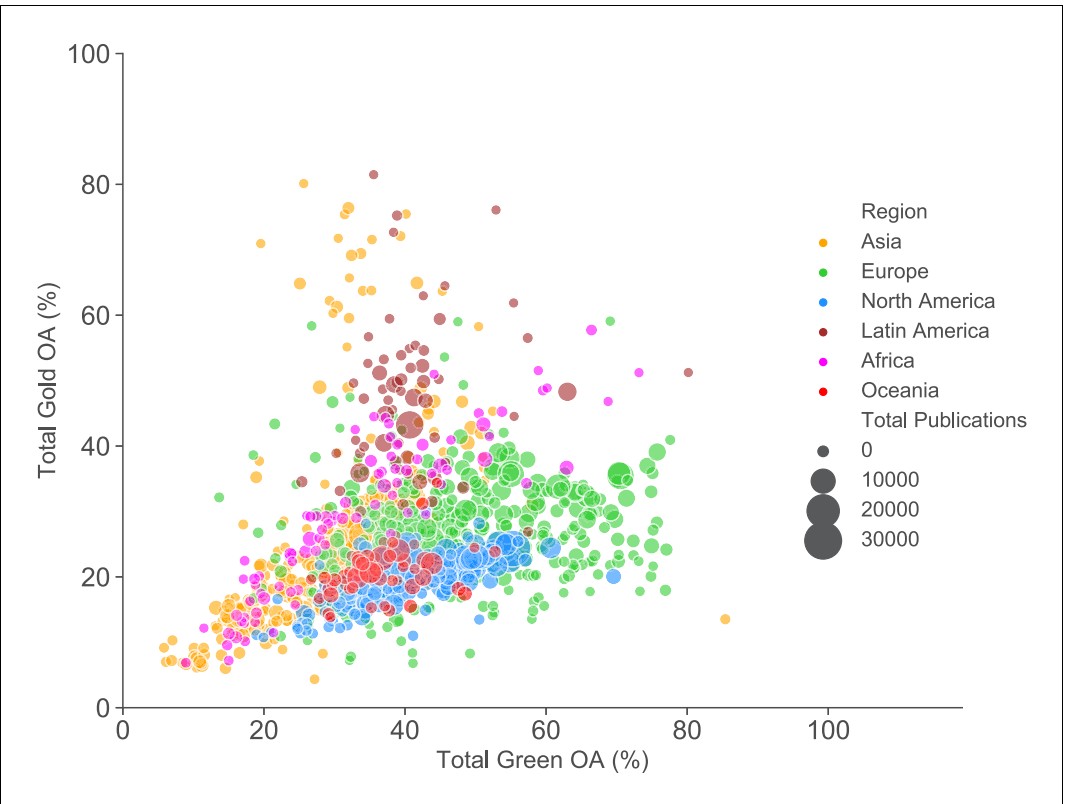

**Figure 3.** Comparing the level of gold and repository-mediated open access of individual universities. Publisher-mediated open access (gold OA) vs repository-mediated open access (green OA) by institution for 2017. Each point plotted is a university, with size indicating the number of outputs analysed and colour showing the region. Articles can be open access through both publisher and repository routes so x and y values do not sum to give total open access. Animated version with figures for each year between 2007 to 2018 can be seen in *Figure 3—animation 1*.

**Figure 3—animation 1.** Comparing the level of gold and repository-mediated open access of individual universities from 2007 to 2018.

https://elifesciences.org/articles/57067#fig3video1

---

figure supplement 3). Amongst countries with a large number of universities in the dataset, the UK is a clear leader with Indonesia, Brazil, Columbia, the Netherlands, and Switzerland showing a strong performance.

To examine the global picture for the 1,207 universities in our dataset and to interrogate different paths to open access, we plot the overall level of repository-mediated ('green') and publisher-mediated ('gold') open access for each university over time coloured by region as previously (*Figure 3*).

Overall universities in Oceania (Australia and New Zealand) and North America (Canada and the US) lag behind comparators in Europe (on repository-mediated open access) and Latin America (on publisher-mediated open access). Asian universities are highly diverse: there are some high performers in the top 100 institutions,

particularly for publisher-mediated open access, but many also lag behind (*Figure 2—figure supplement 2*). Africa is also highly diverse but with a skew towards high performance, with an emphasis on publisher-mediated open access (i.e. gold open access). This may reflect our sampling which is skewed towards institutions with the largest (formally recorded) publishing volumes, many of which receive significant portions of their funding from international donors with strong open access requirements. Latin American institutions show high levels of publisher-mediated open access throughout the period illustrated. This is due to substantial infrastructure investments in systems like SciELO starting in the 1990s.

## Investigating the possible effects of policy interventions

If our goal is to provide data on the effectiveness of interventions then our analysis should be capable of identifying potential effects of policy change. In 2012, the UK Research Councils, following the Finch Report, provided additional funding to individual universities to support open access publishing. The amount of additional funding relates to existing research council funding. In *Figure 4A*, we show the annual change in publisher-mediated open access for three UK universities with the largest additional funding, and three with significantly less additional funding (*Lawson, 2018*). In either 2012 or 2013, a slight increase in publisher-mediated open access across all the universities could be detected. As the additional funding tails off in 2015, the rate of growth falls back. Similarly,

there was an increase in the proportion of hybrid open access publications, which largely require article processing charges, and an increase in proportion of hybrid open access within all publisher-mediated open access around the same period (*Figure 4—figure supplement 1A and B*).

*Figure 4B* shows the growth of content in UK university repositories from 2000 to 2017 compared to two universities from other regions. In 2015, to be included in the UK Research Excellence Framework, universities had to deposit their research outputs in a repository. This policy shift was profound because it relates to an assessment exercise and funding which covers all disciplinary areas, and all universities. The dominance of UK universities in the top 100 for both overall open access and repository-mediated open access, as well as the commitment to

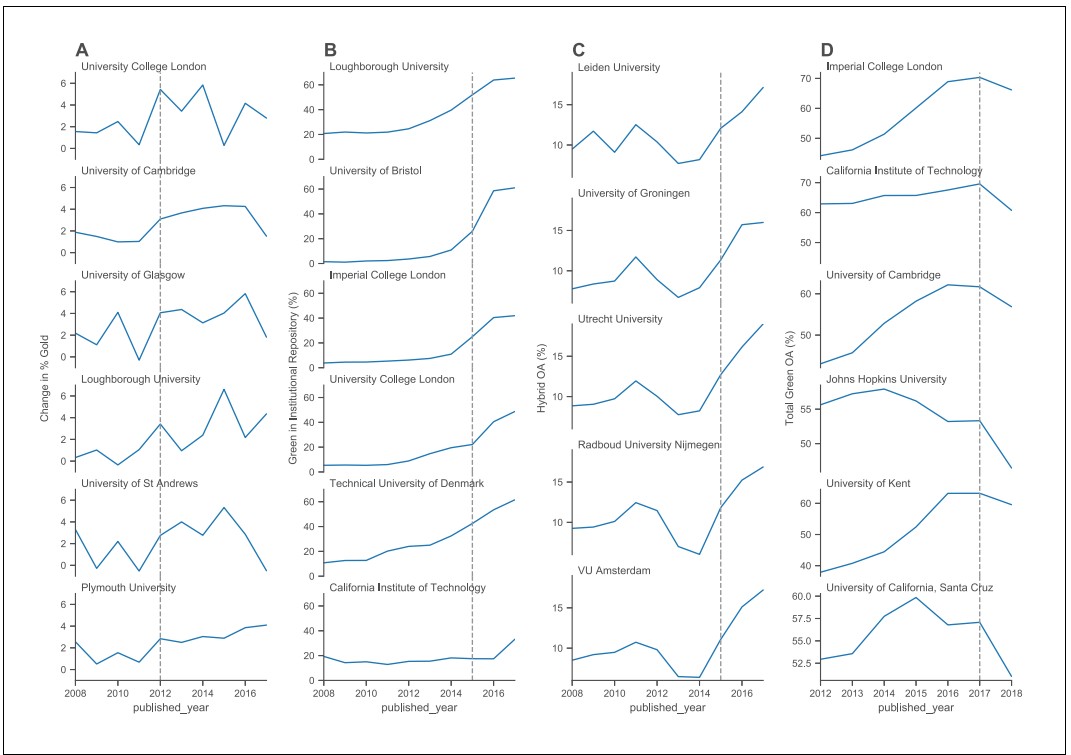

**Figure 4.** Monitoring the effect of policy interventions for selected groups of universities. (A) The annual change in percentage (rolling current year percentage minus the previous year percentage) of gold OA for six UK universities. The top three universities are those with the largest additional funding compared to the bottom three universities which received less additional funding. (B) The annual percentage of green OA through the home institutional repositories of four UK universities compared to high performing universities from elsewhere. (C) The annual percentages of hybrid OA at five universities in the Netherlands. (D) Three pairs of UK and US universities, selected based on having a similar size and level of green OA. The annual percentages of total green OA are depicted for each university. Additional figures are provided in *Figure 4—figure supplement 1*.

The online version of this article includes the following figure supplement(s) for figure 4:

**Figure supplement 1.** Monitoring changes in the percentage of OA publications for selected groups of universities from the UK and the Netherlands.

achieve 100% open access coverage being made by such a large number of universities, is potentially driven in large part by that intervention.

Next, we investigated how the take up of hybrid open access publishing options in the Netherlands were influenced by deals with Springer in 2015, and Wiley in 2016 (*Figure 4C*). These deals essentially allow authors from Dutch universities to publish their work open access in a list of hybrid journals at no cost. We found that across the Netherlands levels of publishing in hybrid open access journals show a sharp turn of increase from 2014 onwards with a less pronounced effect (more smooth increases) for publishing in pure (i.e., gold) open access (compare *Figure 4—figure supplement 1C and D*).

Finally, in *Figure 4D* we show the possible effect of subtle differences in policy relating to acceptable embargo periods. UK research and funding council polices have been aggressive in reducing embargo lengths, mandating six months for STEM subjects and twelve months for humanities and social science (HSS) subjects. The potential effect of embargoes can be seen in the data for repository-mediated open access as a dip in the most recent years of publication. Using Unpaywall data from late 2019, we see a dip in repository-mediated open access performance for UK universities in 2018 but a limited effect on 2017. By comparison with three of the highest performing US universities, comparable in size and overall ranking (see *Figure 2—figure supplement 2*) we see an extended dip in performance, indicative of an acceptance of longer embargoes.

### Different institutional paths towards open access

In *Figures 3* and *4* we see evidence of different paths towards open access, depending on the context and resources. The idea of mapping these paths is shown explicitly for a subset of universities in *Figure 5*. This shows the paths taken by a selection of Latin American institutions and two sets of UK universities over time. For the UK universities shown, three received substantial funding from the UK research councils for open access publishing, whereas the other three received less additional funding and followed an alternate route, emphasising repository-mediated open access.

In contrast, the Latin American institutions already have high levels of publisher-mediated open access at our earliest time point, as discussed earlier. However, our data suggests a fall in overall open access amongst Latin American universities from 2012 onwards, which we ascribe to an increased pressure to publish in 'international' journals that are often subscription based, and for which Latin American scholars are reluctant or unable to pay hybrid article processing charges.

## Discussion

Our results have significant implications for the details of policy interventions. Firstly, we have demonstrated the ability to detect signals of policy interventions in the behaviour of institutions. We see potential effects and results arising from the efforts of national funders and policy makers, particularly in the UK. The combined policy change and funding provided by the UK Research Councils in 2012 is associated with an increase in the level of publisher-mediated open access, and the level of increase appears to be associated with the level of funding provided. Similarly, the requirement for outputs to be deposited in a repository for eligibility for the 2021 Research Excellence Framework is associated with substantial increase in repository-mediated open access around 2015.

These findings may also have implications for deciding on the effectiveness of directly funding open access publishing. It is perhaps surprising to some readers that the overall levels of publisher-mediated open access in the UK are not higher. Specific funders, most notably the Wellcome Trust, have achieved very high levels of open access for articles from research they support through the provision of funding for open access publication. In addition, the UK Research Councils invested significant resources in supporting gold open access. However, these have not translated to high levels of publisher-mediated open access across the full diversity of outputs of UK institutions. The majority gains over the past five years have come from repository-mediated open access.

In the animated version of *Figure 3* (*Figure 3—animation 1*), there is a clear signal of saturation with respect to publisher-mediated open access (gold open access) for European and North American universities. With few exceptions, institutions do not achieve levels of gold open access greater than 40% and this level is stable from 2014 to 2018. Similarly, in *Figure 4* we see evidence of shifts in response to stimuli (funding and policy interventions) which then stabilise. Even those UK universities with very high levels of repository-mediated

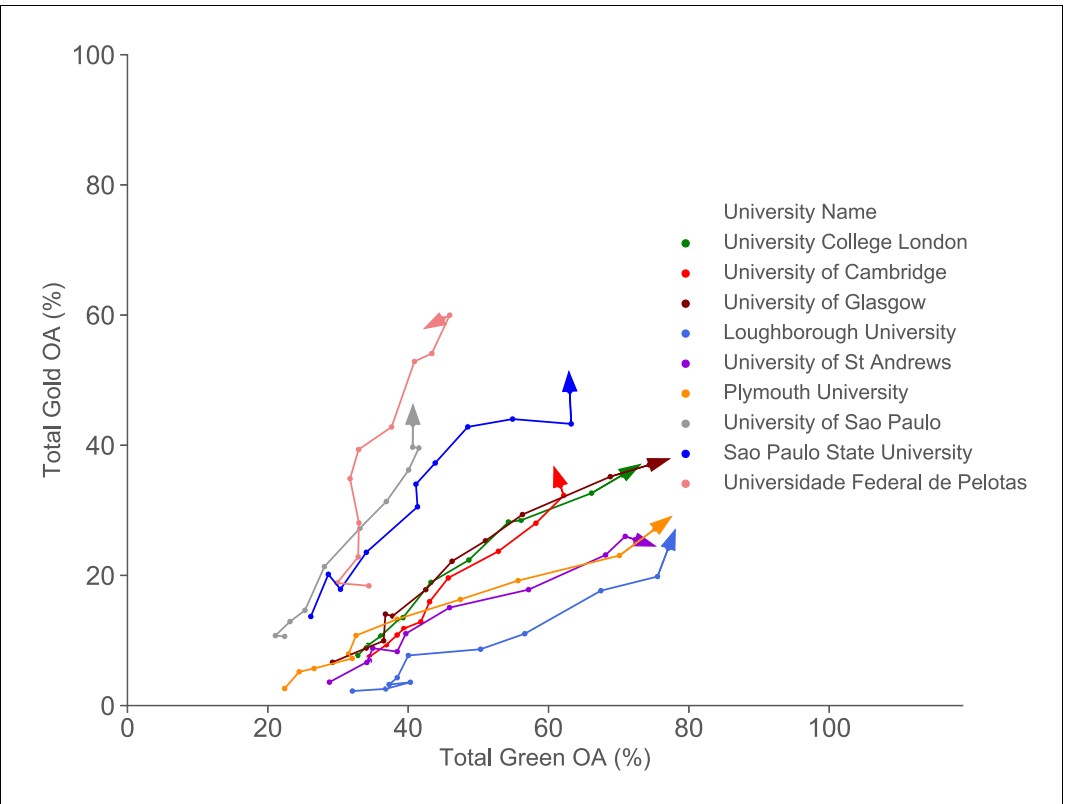

**Figure 5.** Comparing different paths to open access (gold OA versus green OA) for a selected set of universities from 2007 to 2017. This figure compares the different open access routes taken by three groups of universities: three UK universities (University College London, University of Cambridge and University of Glasgow) that received substantial funding for open access publishing (combined gold and green OA increases), three UK universities (Loughborough University, University of St Andrews and Plymouth University) that received less funding (more green OA focused), and three Latin American universities (more gold OA focused). The dots represent the % of total gold OA publications and % of total green OA publications for the specified universities for each year from 2007 to 2017, where the arrow indicates the direction of time.

**Figure 5—animation 1.** An animation comparing different paths to open access (gold OA versus green OA) for a selected set of universities from 2007 to 2017.

https://elifesciences.org/articles/57067#fig5video1

open access (green) see a slowing down of the rise in levels a few years after the Research Excellence Framework policy intervention.

These signals suggest that achieving 100% open access may be very difficult, and possibly expensive to achieve. There will always be areas and cases where open access is challenging. Current challenges include disciplinary areas such as the humanities where suitable business models and venues are still developing, as well as types of content, particularly books, where there is greater overlap between scholarly and general publishing. Both of these could be addressed rapidly through direct funding, but this may not be the most efficient approach. Achieving '100% open access' may therefore require a tighter definition of exactly which

outputs are in scope. For those areas where we see signals of saturation much lower than 100% these are likely signals of the complexity of the system, and of large categories of outputs where open access is harder to achieve, or the motivation of institutions (including authors, libraries, and other support staff) to achieve it is lower.

In comparison to results presented by *Robinson-Garcia et al., 2019*, *Robinson-Garcia et al., 2020*, our analysis shows significantly improved performance of Latin American universities. This is driven by our inclusion of data sources beyond Web of Science, which has resulted in increased publisher-mediated open access for Latin America (*Huang et al., 2020b*). Similarly, our results also depict more clear observations of some high performing African and Asian universities.

These produce a slightly different view in contrast to the one where UK universities dominate the top-ranking positions for open access.

For Latin America, this result indicates how effective infrastructures such as SciELO can be at supporting the uptake of open access practices. In the case of Africa, there may be effects of funder requirements (with funders such as the Bill and Melinda Gates Foundation and Wellcome Trust that have strong open access requirements playing a significant role) as well as disciplinary spread. In both cases we are likely to have a limited view of the full diversity of research outputs due to their poor capture in information systems from the North Atlantic.

The continued leadership of Latin American institutions on publisher-mediated open access levels is the continuation of a trend set more than a decade ago through the provision of publishing infrastructures. Taken alongside the clear response in the Netherlands for hybrid open access in response to publish and read agreements, this suggests that increasing levels of open access publishing through article processing charges is potentially expensive compared to the costs of providing infrastructure.

Another interesting natural experiment is how the strength of funder actions is associated with overall change in levels of open access. In the Netherlands and the UK in particular, but also in the US, where funder policies have moved from encouragement, to mandates, to monitoring with sanctions for non-compliance, there are substantial shifts in overall levels of open access. By contrast, in countries where policy remains effectively at the level of a recommendation, such as Australia, levels of open access lag significantly. Recent increases in reporting requirements by Australian funders might therefore be expected to lead to a detectable signal over the next 12–24 months.

The value of analysis at the level of universities is that we gain a picture of open access performance across a diverse research ecosystem. We see differences across countries and regions, and differences between universities within countries. Overall, we see that there are multiple different paths towards improving access, and that different paths may be more or less appropriate in different contexts. Most importantly, while further research is needed to unpick the details of the differences in open access provision, we hope this work provides a framework that enables this longitudinal analysis to be taken forward and used wherever it is needed.

## Limitations and further work

Our analysis process includes automated approaches for collecting the outputs related to specific universities, and the analysis of those outputs. Currently the addition of new universities, and the updating of large data sources is partly manual (*Supplementary file 1*), but we also expect to automate this in the near future. As a result, our analysis has limitations in its capacity to provide comparable estimates of open access status across all universities, but does provide a reproducible and transparent view of overall global performance.

There are challenges to be addressed with respect to small universities and research organisations, and we have taken a necessarily subjective view of which institutions to include (see *Supplementary file 1*). Our approach systematically leaves out universities with very small numbers of outputs (i.e., less than 100 outputs), and universities with very extreme open access proportions as these are the universities for which we have less statistical confidence in the results. This is also in-line with our intended focus on research-intensive universities. These small institutions are of significant interest but will require a different analysis approach. We include a country-level summary result for the full set of institutions (including small institutions left out from the main article) in our data set in *Figure 2—figure supplement 3*. We recognise that the inclusion of universities beyond our initial selection can also potentially change the results.

We have used multiple sources of bibliographic information with the goal of gaining a more inclusive view of research outputs. Despite this, there are still limitations in the coverage of these data sources, and a likely bias towards science, technology, engineering and maths (STEM) disciplines. In addition, the focus of Unpaywall on analysis of outputs with Crossref DOIs means that we are missing outputs for disciplines (humanities) and output types (books) where the use of DOIs is lower. While we have performed additional sensitivity analysis on the use of different data sources (*Huang et al., 2020b*), further work is required to understand the relationships between disciplinary biases in data sources and the influence of external policies (such as those of the National Institutes of Health and the Wellcome Trust) on repositories. In addition, due to the nature of this work and to limitations on the use of Web of Science and Scopus APIs, we have collected data from these two sources over a period of time. Although we

expect such changes to be small, those effects are not clearly represented in our data. For other data sources we are able to precisely define the data dump used for our analysis, supporting reproducibility as well as modified analyses. We also recognise that our longitudinal analysis is done post hoc from a fixed time point (i.e., looking back). Our results for a given year of publication will not be the same as the levels of open access that would have been observed *in* that year, due to embargoes and moving paywalls.

In this article, we have drawn comparisons between publisher-mediated open access and repository-mediated open access to identify different institutional paths to open access. The overall trends are retained if we separate out outputs that are only available through repositories (i.e., not free to access via publishers) and compare these to publisher-mediated open access instead. However, we recognise that at the individual university level, this can be dependent on institutional policy and whether the corresponding data can be properly captured by current systems that determines open access status.

There is significant opportunity for improving the data sources on sets of outputs and how they can be grouped (e.g. by people, discipline, organisation, country, etc.). Improvements to institutional identifier systems such as the Research Organisation Registry, increased completeness of metadata records, particularly that provided by publishers via Crossref on affiliation, ORCIDs and funders, and enhancing the coverage of open access status data (for instance by incorporating data from CORE and BASE), will all enhance coverage. There are also opportunities to expand the coverage by incorporating a wider range of bibliographic data sources.

We have sought to make our methodology and approach as reproducible and reusable as is practicable. The main challenges lie in the level of accessibility to the closed data sources (i.e., Web of Science and Scopus) and more generally with data sources that are not available in the form of identifiable snapshots (see *Supplementary file 1*). In addition, semi-automated processing and manual searches are required to link institutional identifiers across multiple data sources (as discussed in detail in *Supplementary file 1*). For the rest of the framework, we have provided transparent descriptions and source code that allow reprocessing, as described in the next section. Further work is required to provide a complete and deployable framework for replication but we are making progress in this area (see https://github.com/The-Academic-Observatory/observatory-platform for an example).

## Acknowledgements

This work was funded by the Research Office of Curtin University through a strategic grant, the Curtin University Faculty of Humanities, and the School of Media, Creative Arts and Social Inquiry. The authors would like to thank the editors and reviewers for their valuable feedback that have helped to improve this article.

**Chun-Kai (Karl) Huang** is in the Centre for Culture and Technology, School of Media, Creative Arts and Social Inquiry, Curtin University, Perth, Australia
karl.huang@curtin.edu.au
https://orcid.org/0000-0002-9656-5932

**Cameron Neylon** is in the Centre for Culture and Technology, School of Media, Creative Arts and Social Inquiry, and Curtin Institute for Computation, Curtin University, Perth, Australia
https://orcid.org/0000-0002-0068-716X

**Richard Hosking** is in the Curtin Institute for Computation, Curtin University, Perth, Australia

**Lucy Montgomery** is in the Centre for Culture and Technology, School of Media, Creative Arts and Social Inquiry, and Curtin Institute for Computation, Curtin University, Perth, Australia
https://orcid.org/0000-0001-6551-8140

**Katie S Wilson** is in the Centre for Culture and Technology, School of Media, Creative Arts and Social Inquiry, Curtin University, Perth, Australia
https://orcid.org/0000-0001-8705-1027

**Alkim Ozaygen** is in the Centre for Culture and Technology, School of Media, Creative Arts and Social Inquiry, Curtin University, Perth, Australia
https://orcid.org/0000-0001-6813-8362

**Chloe Brookes-Kenworthy** is in the Centre for Culture and Technology, School of Media, Creative Arts and Social Inquiry, Curtin University, Perth, Australia

*Author contributions:* Chun-Kai (Karl) Huang, Conceptualization, Resources, Data curation, Software, Formal analysis, Validation, Investigation, Visualization, Methodology, Writing - original draft, Writing - review and editing; Cameron Neylon, Conceptualization, Resources, Data curation, Software, Formal analysis, Funding acquisition, Validation, Investigation, Visualization, Methodology, Writing - original draft, Project administration, Writing - review and editing; Richard Hosking, Resources, Data curation, Software, Writing - review and editing; Lucy Montgomery, Conceptualization, Funding acquisition, Validation, Project administration, Writing - review and editing; Katie S Wilson, Alkim Ozaygen, Chloe Brookes-Kenworthy, Validation, Writing - review and editing

*Competing interests:* The authors declare that no competing interests exist.

**Funding**

| Funder | Author |
|---|---|
| Curtin University of Technology | Chun-Kai Huang Cameron Neylon Richard Hosking Lucy Montgomery Katie S Wilson Alkim Ozaygen Chloe Brookes-Kenworthy |

The funders had no role in study design, data collection and interpretation, or the decision to submit the work for publication.

### Decision letter and Author response

Decision letter https://doi.org/10.7554/eLife.57067.sa1
Author response https://doi.org/10.7554/eLife.57067.sa2

## Additional files

### Supplementary files

• Supplementary file 1. Further detail on the methodology and data sources used to evaluate the open access performance of individual institutions.

• Transparent reporting form

### Data availability

Our technical infrastructure is constructed based on the aim to make openly available both the data and analysis code as much as possible. The derived datasets for analysis, visualisation codes, and the SQL queries used to generate the derived datasets are all made available online as described in Supplementary file 1. Raw data is not provided to preserve the anonymity of institutions and respect the terms of service of data providers.

The following datasets were generated:

| Author(s) | Year | Dataset URL | Database and Identifier |
|---|---|---|---|
| Huang C-K, Neylon C, Hosking R, Montgomery L, Wilson K, Ozaygen A, Brookes-Kenworthy C | 2020 | https://doi.org/10.5281/zenodo.3693221 | Zenodo, 10.5281/zenodo.3693221 |
| Huang C-K, Neylon C, Hosking R, Montgomery L, Wilson K, Ozaygen A, Brookes-Kenworthy C | 2020 | https://doi.org/10.5281/zenodo.3716063 | Zenodo, 10.5281/zenodo.3716063 |

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
