## [Decision Letter]

Thank you for submitting your article "Evaluating institutional open access performance: Methodology, challenges and assessment" for consideration by *eLife*. Your article has been reviewed by 2 peer reviewers, and the evaluation has been overseen by two members of the *eLife* Features Team. The following individuals involved in review of your submission have agreed to reveal their identity: Bianca Kramer (Reviewer #1).

The reviewers have discussed the reviews with one another and the Reviewing Editor has drafted this decision to help you prepare a revised submission.

Summary:

This work provides a re-usable framework for tracking the open access performance of institutions and identifies instances where funding and policy interventions have changed the level of open access happening at the institutional, country or regional level.

Major comments to address:

1) As Figure 2 will be difficult to see and interpret in the web and PDF versions of the article, please can you replace this figure with Supplementary Figure 6. This will also be more in line with the examples given in the paper highlighting potential uses of the framework to analyse policy effects which are based on either comparing institutional results at country or region level for the full dataset (Figure 3), or on comparing specific institutions based on a common or differing policy context (Figure 4,5). Furthermore, including the ranking of top 100 institutions in terms of the proportion of different types of OA as a main result could invite undue focus on comparing individual institutions in a competitive way, which seems not in line with the aims of the work.

2) The paper introduces several analytical categories that it states should be used as the 'minimum reporting standard'. However, why these categories should be the 'minimum reporting standard' requires a much stronger justification. For example, the authors have included data sources with restricted access (Scopus, WoS) along with the Microsoft Academic Graph, to increase recall. Additionally, there is no mention as to why research disciplines as a possible determinant of OA publication are not included as a "minimum reporting standard". It is also unclear who the minimum reporting standard is for - Universities? Ranking producers? Researchers?

Overall, this is a pretty good paper with good data and methods, but I do not think that it is a good proposal of a "minimum reporting standard" and suggest this claim is dropped from the paper. The sentence "this study proposes a minimum reporting standard for institutional open access evaluation" should be removed from Section 1.1 and other places where this is mentioned in the paper.

3) The following points should be added to the Discussion or Limitations section:

- the effects and limitations of comparing only 'all green OA' (and not 'green only')

to 'gold OA', as 'all green' reflects both the extent to which gold OA papers are also included in repositories, as the extent to which otherwise closed papers are made available in repositories, and this may differ for different universities, depending on institutional policies.

- the limitation of longitudinal analysis done from a fixed point in time (i.e. looking back), as for earlier years, the effect of embargoes (in proportion of green OA) and moving walls (for bronze OA) will influence the results - i.e. results of the previous year are not just a reflection of proportion of OA in that year.

- the effect of disciplinary differences, regarding a) the extent to which the research outputs included (present in WoS/Scopus/MA, with DOI) can be expected to represent the total research output of an institution, depending on b) the extent to which observed levels of green OA are influenced by external policies such as those of NIH and Wellcome for biomedical research, resulting in green OA through PubMedCentral and EuropePMC, rather than through home repositories.

- the potential of extension of the approach to universities beyond the initial selection (which favours research-intensive universities), and expected differences in results.

4) It is stated in Section 1.1 that the authors intend to favour recall over precision. From looking at the details provided in the supplementary methods, it seems that what the authors mean here is that by using a variety of data sources they aimed to increase recall (but not at the expense of precision). If this is the case, please can the authors revise this sentence. If this is not a mistake, a justification as to why it makes more sense to focus on more data rather than better data needs to be included, especially when the next point on the list focuses on size-independent indicators, for which precision is more important than recall.

5) To give the table displaying the different OA categories more context, please can you add a caption similar to the information given in the Supplementary Methodology. This should include, which OA types are reported in this paper (Total OA, Gold, Hyrbid, Green, and Green in Homo Repo) and an explanation as to why you chose to include these categories in the main analysis and not the other types (Hybrid, Bronze, Gold DOAJ and Green only).

6) I think that shifts in trends that coincide with policy changes should be included in whatever body of evidence used to determine whether policies have been effective. That said, I think that the empirical setup simply doesn't allow to make any "effect" claims, and I would advise the authors to use the term sparingly and to speak perhaps of "possible effect" or something like that. For example, I am not convinced by the interpretation of Figure 4A: the trends could very well be a somewhat random fluctuation in the 2%-6% range, and I think it is a bit exaggerated to claim that this figure shows any kind of clear effects.

7) In the accompanying sensitivity analysis, some interesting patterns are detected (higher levels of total OA and gold OA, but lower levels of green OA for Latin America when Microsoft Academic (MA) is included, and lower levels of total OA and green OA for the UK when MA is included (cf. WoS and/or Scopus only). It would be good to provide some context/explanations for these observed effects, either in the white paper or in the main paper itself.

8) Consider adding a paragraph on the possibilities and limitations of re-using this methodology, particularly in regards to the closed data sources (with ramifications for what data can be shared). It may also be worth including more detail on the semi-automatic steps of the methodology so that people would be able to reproduce them if they have access to the data sources. In a broader context, there might be an opportunity to discuss the potential of making the framework more widely available to interested parties to create an open infrastructure that can assess open access levels at various aggregation levels.

Data/Stats to be included:

1) Include an explanation of how the confidence intervals used are derived (esp. since they are an important addition of this analysis compared to previous analyses of OA levels).

2) Either provide anonymized or de-anonymized data for all universities in the sample, not just for a subset.

3) Please can you make the examples of code in the supplementary methodology available to download.

---

## [Author Response]

Major comments to address:1) As Figure 2 will be difficult to see and interpret in the web and PDF versions of the article, please can you replace this figure with Supplementary Figure 6. This will also be more in line with the examples given in the paper highlighting potential uses of the framework to analyse policy effects which are based on either comparing institutional results at country or region level for the full dataset (Figure 3), or on comparing specific institutions based on a common or differing policy context (Figure 4,5). Furthermore, including the ranking of top 100 institutions in terms of the proportion of different types of OA as a main result could invite undue focus on comparing individual institutions in a competitive way, which seems not in line with the aims of the work.

As suggested in the review report, the Figure showing the top 100 is now relocated to Figure 2 – figure supplement 2. Figure 2 is replaced with the figure showing groupings by regions for 2017, with figures for 2016 and 2018 included in Figure 2 – figure supplement 1.

2) The paper introduces several analytical categories that it states should be used as the 'minimum reporting standard'. However, why these categories should be the 'minimum reporting standard' requires a much stronger justification. For example, the authors have included data sources with restricted access (Scopus, WoS) along with the Microsoft Academic Graph, to increase recall. Additionally, there is no mention as to why research disciplines as a possible determinant of OA publication are not included as a "minimum reporting standard". It is also unclear who the minimum reporting standard is for - Universities? Ranking producers? Researchers?Overall, this is a pretty good paper with good data and methods, but I do not think that it is a good proposal of a "minimum reporting standard" and suggest this claim is dropped from the paper. The sentence "this study proposes a minimum reporting standard for institutional open access evaluation" should be removed from Section 1.1 and other places where this is mentioned in the paper.

We have now removed the phrase “minimum reporting standard”. We have also updated the discussion to include ways to which our methodology can be adapted to perform analysis for other categories, such as disciplines, individuals, countries, etc, instead of institutions.

3) The following points should be added to the Discussion or Limitations section:i) the effects and limitations of comparing only 'all green OA' (and not 'green only')to 'gold OA', as 'all green' reflects both the extent to which gold OA papers are also included in repositories, as the extent to which otherwise closed papers are made available in repositories, and this may differ for different universities, depending on institutional policies.

We have included an extra paragraph in the Limitations section to discuss this issue. The main message is that the general trends are not affected, but we recognise that at the individual university level this can be dependent on institutional policy and data quality.

ii) the limitation of longitudinal analysis done from a fixed point in time (i.e. looking back), as for earlier years, the effect of embargoes (in proportion of green OA) and moving walls (for bronze OA) will influence the results - i.e. results of the previous year are not just a reflection of proportion of OA in that year.

This is now included in the Limitations section.

iii) The effect of disciplinary differences, regarding a) the extent to which the research outputs included (present in WoS/Scopus/MA, with DOI) can be expected to represent the total research output of an institution, depending on b) the extent to which observed levels of green OA are influenced by external policies such as those of NIH and Wellcome for biomedical research, resulting in green OA through PubMedCentral and EuropePMC, rather than through home repositories.

Additional discussion on this matter is now included in the Limitations section.

iv) The potential of extension of the approach to universities beyond the initial selection (which favours research-intensive universities), and expected differences in results.

This is now mentioned in the Limitations section.

4) It is stated in Section 1.1 that the authors intend to favour recall over precision. From looking at the details provided in the supplementary methods, it seems that what the authors mean here is that by using a variety of data sources they aimed to increase recall (but not at the expense of precision). If this is the case, please can the authors revise this sentence. If this is not a mistake, a justification as to why it makes more sense to focus on more data rather than better data needs to be included, especially when the next point on the list focuses on size-independent indicators, for which precision is more important than recall.

This has been corrected to say “increase recall but not at the expense of precision”.

5) To give the table displaying the different OA categories more context, please can you add a caption similar to the information given in the Supplementary Methodology. This should include, which OA types are reported in this paper (Total OA, Gold, Hyrbid, Green, and Green in Homo Repo) and an explanation as to why you chose to include these categories in the main analysis and not the other types (Hybrid, Bronze, Gold DOAJ and Green only).

This is now included under the Methods section, below the table of OA definitions.

6) I think that shifts in trends that coincide with policy changes should be included in whatever body of evidence used to determine whether policies have been effective. That said, I think that the empirical setup simply doesn't allow to make any "effect" claims, and I would advise the authors to use the term sparingly and to speak perhaps of "possible effect" or something like that. For example, I am not convinced by the interpretation of Figure 4A: the trends could very well be a somewhat random fluctuation in the 2%-6% range, and I think it is a bit exaggerated to claim that this figure shows any kind of clear effects.

We have now rephrased the wording used in response to this comment.

7) In the accompanying sensitivity analysis, some interesting patterns are detected (higher levels of total OA and gold OA, but lower levels of green OA for Latin America when Microsoft Academic (MA) is included, and lower levels of total OA and green OA for the UK when MA is included (cf. WoS and/or Scopus only). It would be good to provide some context/explanations for these observed effects, either in the white paper or in the main paper itself.

Additional discussions have been added to the white paper on sensitivity analysis.

8) Consider adding a paragraph on the possibilities and limitations of re-using this methodology, particularly in regards to the closed data sources (with ramifications for what data can be shared). It may also be worth including more detail on the semi-automatic steps of the methodology so that people would be able to reproduce them if they have access to the data sources. In a broader context, there might be an opportunity to discuss the potential of making the framework more widely available to interested parties to create an open infrastructure that can assess open access levels at various aggregation levels.

A short description is now included at the end of the “Limitations” section to discuss limitations of reuse and further work in this direction.

Data/stats to be included:1) Include an explanation of how the confidence intervals used are derived (esp. since they are an important addition of this analysis compared to previous analyses of OA levels).

Additional explanations are included in Supplementary file 1, under the section “Description of data workflow and selection criteria”.

2) Either provide anonymized or de-anonymized data for all universities in the sample, not just for a subset.

We would like to keep universities anonymised except for the top 100, which highlight important messages about universities not traditionally ranked the highest leading the way for OA. As such we have included two different sets of the processed data. The first includes all universities in anonymised form, and the second with only the top 100 universities included in de-anonymised form. This is made clearer at the end of the “Methods” section.

3) Please can you make the examples of code in the supplementary methodology available to download.

These are available via Zenodo, with the link provided in the article and Supplementary file 1.